# Osteosarcoma Arising in Fibrous Dysplasia of the Long Bone: Characteristic Images and Molecular Profiles

**DOI:** 10.3390/diagnostics12071622

**Published:** 2022-07-04

**Authors:** Han Gyeol Kim, Jong Hun Baek, Kiyong Na

**Affiliations:** 1Department of Pathology and Translational Genomics, Samsung Medical Center, Sungkyunkwan University School of Medicine, Seoul 06351, Korea; hangyeol_kim@naver.com; 2Department of Orthopaedic Surgery, Kyung Hee University Hospital, Kyung Hee University College of Medicine, Seoul 02447, Korea; paeton81@khu.ac.kr; 3Department of Pathology, Kyung Hee University Hospital, Kyung Hee University College of Medicine, Seoul 02447, Korea

**Keywords:** osteosarcoma, fibrous dysplasia, malignant transformation, next-generation sequencing

## Abstract

Fibrous dysplasia (FD) is a benign fibro-osseous lesion that frequently involves the craniofacial bones and femur. Malignant transformation of FD is a rare occurrence. We report a 38-year-old woman with osteosarcoma (OS) arising from FD of the femur. Magnetic resonance imaging revealed a well-defined lesion in the medulla of the femur, with cortical thinning and local bone destruction. Wide excision of the femur was performed. Grossly, the inner part of the mass was hard and tan-gray in color, and the outer part of the mass adjacent to the cortex showed myxoid discoloration with infiltrative borders. Microscopically, most of the tumor consisted of curvilinear woven bone and fibrous stroma with bland spindle cells, which transitioned to the outer portion of the tumor, showing cellular proliferation of pleomorphic cells with frequent mitotic activity. Next-generation sequencing revealed *GNAS* and *TP53* mutations, and the diagnosis of OS arising from FD was strongly supported. This case highlights the characteristic images and molecular features of the malignant transformation of FD.

**Figure 1 diagnostics-12-01622-f001:**
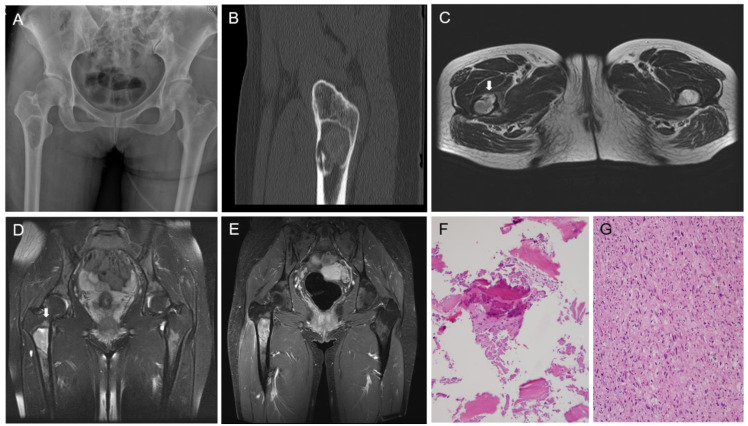
Fibrous dysplasia (FD) is a benign fibro-osseous lesion with varying proportions of fibrous and osseous components. Depending on whether FD affects one or multiple bones, it is divided into monostotic and polyostotic types [1]. The craniofacial bones and femur are commonly involved in monostotic FD [2]. Malignant transformation of FD rarely occurs, and osteosarcoma (OS) is the most common histological subtype [3,4]. To date, there have been few reports demonstrating the molecular profiles of OS in FD [5]. Herein, we describe the clinicopathological, immunohistochemical, and molecular features of OS arising from FD of the proximal femur in a middle-aged woman. A detailed case presentation is presented in Figure 1 and Figure 2. As the degree of nuclear atypia and tumor cellularity in the outer portion suggested high-grade OS, we considered dedifferentiated low-grade central osteosarcoma (LGCOS) and OS arising in FD as the main differential diagnoses. LGCOS can be differentiated from FD by the presence of the following histologic features: infiltrative growth, cellularity, cytologic atypia, and mitotic activity [6,7]. Infiltrative growth, such as permeation or extension to the cortical bone and adjacent soft tissue, is an important gross feature of LGCOS. Microscopically, increased cellularity and intersecting fascicular patterns of spindle cells are usually detected in LGCOS but not in FD. LGCOS shows focal nuclear atypia, including hyperchromatic, enlarged nuclei, and irregular nuclear membranes. In addition to the morphological features, investigation of the underlying molecular profiles is helpful for differential diagnosis. While FD frequently harbors *GNAS* mutations [8,9], LGCOS frequently shows MDM2 and/or CDK4 amplification [10]. We conducted next-generation sequencing (NGS) analysis to determine whether the high-grade OS was dedifferentiation from LGCOS or malignant transformation from FD. DNA sequence-change analysis revealed *GNAS* (p.R201H, c.602G > A) and *TP53* mutations (p.W53Cfs*70, c.159delG). Copy number variation analysis revealed a gain of *PDGFRA*, *KIT*, and *CTNND2* and a loss of *CDKN2A*. *TP53* mutations and *CDKN2A* deletions have been reported in primary OS or secondary OS associated with FD [5,11]. Our NGS results strongly supported the diagnosis of malignant transformation of FD. Following surgical resection and pathological diagnosis, the patient underwent Adriamycin-based chemotherapy and was alive without evidence of disease recurrence for 6 months. This case highlights the macroscopic, microscopic, and molecular characteristics of the malignant transformation of FD into OS. The diagnostic approaches, characteristic images, and interpretations of ancillary tests demonstrated in this case would help pathologists and clinicians make accurate diagnoses of malignant transformation of FD and their differential diagnosis from dedifferentiated LGCOS, which is essential for establishing an optimal treatment plan. A 38-year-old woman with no significant medical history was referred with stabbing pain in right hip joint. The pain occurred during load-bearing activities such as walking for 4 months, and became worse even after taking medicine. (**A**) X-ray shows a well-defined intramedullary lesion, 5.6 × 3.3 × 2.5 cm in size. (**B**) CT scan highlighted cortical destruction. (**C**) T2-weighted axial MR image reveals focal cortical destruction with thinning and endosteal erosion (arrow) in the anterior aspect of the femur. (**D**) Fat-saturated T2-weighted coronal MR image reveals a well-defined intramedullary lesion with peritumoral bone edema (arrow) in the meta-diaphysis of the femur. (**E**) Gadolinium-enhanced T1-weighted coronal MR image reveals heterogenous enhancement. (**F**) CT-guided bone biopsy was performed because pain and radiographic finding of cortical thinning were atypical features for fibrous dysplasia and suggested the possibility of malignancy. The first biopsy specimen includes mainly blood clots with some bland-looking spindle cells and woven bone. (**G**) The second biopsy reveals atypical spindle cells with moderate-to-high cellularity and frequent mitotic figures.

**Figure 2 diagnostics-12-01622-f002:**
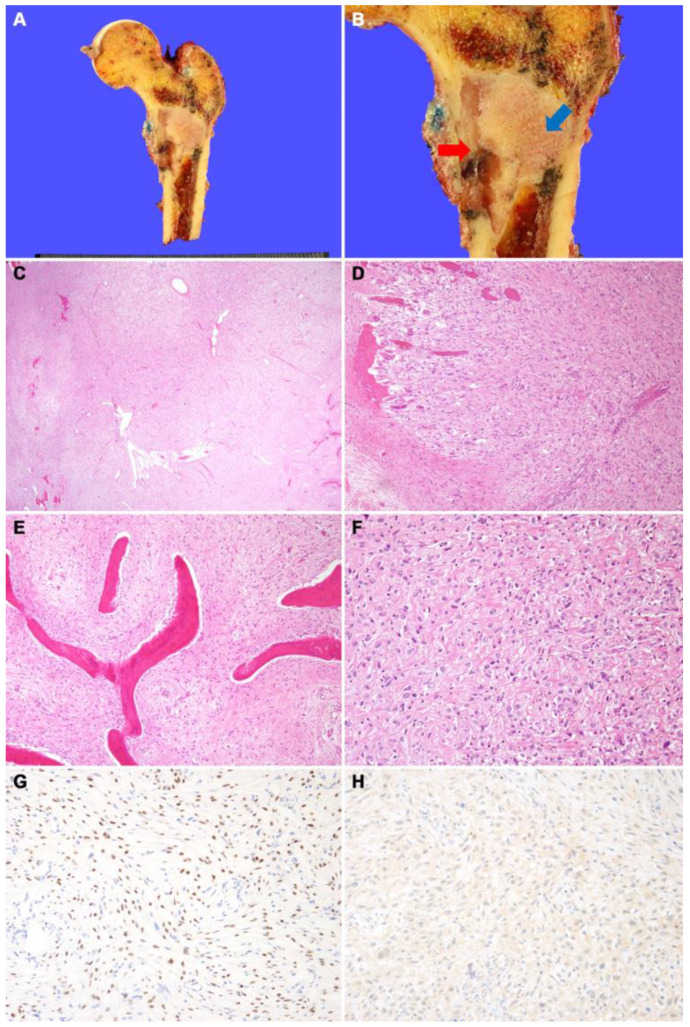
(**A**) Wide excision of proximal femur was performed. (**B**) Grossly, fibrous dysplasia portion is seen as a well-demarcated tan-gray mass with hard texture in the metaphysis of the femur (blue arrow). The osteosarcoma portion of the mass shows gray myxoid discoloration and fragile texture (red arrow). This portion was contiguous with cortical bone destruction and periosteal soft-tissue extension. (**C**) On low magnification, the tumors consist of an inner loosely cellular portion containing woven bone (right) and an outer highly cellular portion without woven bone (left). (**D**) The tumors destroyed the cortical bone and extended into the skeletal muscle. (**E**) On high magnification, the inner part of the tumor consists of curvilinear woven bone and fibrous stroma with bland spindle cells. (**F**) The outer part of the tumor shows highly cellular proliferation of pleomorphic cells with irregularly shaped hyperchromatic nuclei. Malignant osteoid interlaced the tumor cells. (**G**) SATB2 immunostaining demonstrates diffuse nuclear positivity. (**H**) CDK4 immunostaining shows negative result, suggesting a lack of *MDM2* and/or *CDK4* amplification.

## Data Availability

The data presented in this study are available upon request from the corresponding author. The data are not publicly available because of privacy and ethical restrictions.

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
