# Peer review of "Osteosarcoma Arising in Fibrous Dysplasia of the Long Bone: Characteristic Images and Molecular Profiles"

_diagnostics, 2022, doi:10.3390/diagnostics12071622_

Round 1

Reviewer 1 Report

This is a nicely written case report which describes a case of osteosarcoma (OS) arising in fibrous dysplasia (FD) of the femur.

The authors also include a brief discussion of the differential diagnosis between low-grade central osteosarcoma (LGCOS) and OS arising in FD and molecular analysis of next generation sequencing (NGS) which can support proposed pathogenetic mechanisms of malignant progression in these tumors.

This would be a worthwhile contribution to the interesting images (case report) section of the Journal given that only few cases of transformed OS arising in the FD of the long bone have been reported in the literature.

I recommend several minor revisions.

I suggest the following points to help improve the manuscript prior to publication:

Point 1.

2nd paragraph of the main text: “A detailed case presentation is presented in Figures 1 and 2. As the degree of nuclear atypia and tumor cellularity in outer portion suggested high-grade OS, we considered dedifferentiated LGCOS and OS arising in FD as the main differential diagnoses”:

Please spell out “LGCOS” because the first time you mention a phrase that can be abbreviated, you should spell it out in full and provide the abbreviation in parentheses.

Point 2.

Figure 2B legend:

Suggest indicating transformed osteosarcoma (OS) area and fibrous dysplasia (FD) area for clarity.

Point 3.

4th paragraph of the main text: “This case highlights the macroscopic, microscopic, and molecular characteristics of 58 the transformation of FD into OS”:

Suggest “malignant transformation” instead of “transformation” for clarity.

Author Response

Reviewer 1

Comments and Suggestions for Authors

This is a nicely written case report which describes a case of osteosarcoma (OS) arising in fibrous dysplasia (FD) of the femur. The authors also include a brief discussion of the differential diagnosis between low-grade central osteosarcoma (LGCOS) and OS arising in FD and molecular analysis of next generation sequencing (NGS) which can support proposed pathogenetic mechanisms of malignant progression in these tumors. This would be a worthwhile contribution to the interesting images (case report) section of the Journal given that only few cases of transformed OS arising in the FD of the long bone have been reported in the literature. I recommend several minor revisions. I suggest the following points to help improve the manuscript prior to publication:

Point 1.

2nd paragraph of the main text: “A detailed case presentation is presented in Figures 1 and 2. As the degree of nuclear atypia and tumor cellularity in outer portion suggested high-grade OS, we considered dedifferentiated LGCOS and OS arising in FD as the main differential diagnoses”:

Please spell out “LGCOS” because the first time you mention a phrase that can be abbreviated, you should spell it out in full and provide the abbreviation in parentheses.

Answer) We have corrected as indicated. 

Point 2.

Figure 2B legend: Suggest indicating transformed osteosarcoma (OS) area and fibrous dysplasia (FD) area for clarity.

Answer) We have corrected as suggested.

Point 3.

4th paragraph of the main text: “This case highlights the macroscopic, microscopic, and molecular characteristics of 58 the transformation of FD into OS”:

Suggest “malignant transformation” instead of “transformation” for clarity.

Answer) We have corrected as suggested.

Reviewer 2 Report

The radiological part needs to be improved:

- You stated that 'cortical thinning with local cortical bone destruction' was present, please better show this with a magnification of images (X-ray and/or MRI).

- Moreover, was peritumoral bone oedema present? Do you have a fluid-sensitive sequence such as STIR or T2w-fat sat?. If yes please add this to the paper.

- Was contrast injection performed on MRI study? was CT performed?

- Which clinical or imaging features led to histological examination in this well-defined lesion? This is very interesting in clinical practice. (Pain and cortical thinning?)

Author Response

Reviewer 2

Comments and Suggestions for Authors

The radiological part needs to be improved:

- You stated that 'cortical thinning with local cortical bone destruction' was present, please better show this with a magnification of images (X-ray and/or MRI).

Answer) We have change Figure 1 for better demonstration of the lesion.

- Moreover, was peritumoral bone oedema present? Do you have a fluid-sensitive sequence such as STIR or T2w-fat sat?. If yes please add this to the paper.

Answer) We added MRI T2 fat sat image as you recommended.

- Was contrast injection performed on MRI study? was CT performed?

Answer) We added CT image and gadolinium enhanced MR image.

- Which clinical or imaging features led to histological examination in this well-defined lesion? This is very interesting in clinical practice. (Pain and cortical thinning?)

Answer) We performed biopsy because both clinical symptom (pain) and radiographic finding (cortical thinning) were atypical features that were different from those observed in fibrous dysplasia.

Round 2

Reviewer 2 Report

I appreciate the revisions performed.

The radiologic composed figure seems to be rotated, please adjust.

You stated that the surgical procedure was performed because of pain and radiological signs of cortical interruption; please add this information into the manuscript.

Author Response

I appreciate the revisions performed.

The radiologic composed figure seems to be rotated, please adjust.

You stated that the surgical procedure was performed because of pain and radiological signs of cortical interruption; please add this information into the manuscript.

Answer) We have made the correction as suggested,  figure rotation and description of biopsy process in Figure 1 and its legends.

Thank you for sincere review.